# Potential Pharmacokinetic Interactions of Common Cardiovascular Drugs and Selected European and Latin American Herbal Medicines: A Scoping Review

**DOI:** 10.3390/plants12030623

**Published:** 2023-01-31

**Authors:** Jose M. Prieto-Garcia, Louise Graham, Osamah Alkhabbaz, Andre L. D. A. Mazzari

**Affiliations:** 1Centre for Natural Products Discovery, School of Pharmacy and Biomolecular Sciences, Liverpool John Moores University, Liverpool L3 3AF, UK; 2School of Pharmacy, University College London, London WC1N 1AX, UK

**Keywords:** Brazil, Europe, herb–drug interactions, cardiovascular, hypertension, hyperlipidemia

## Abstract

Background: Herb–drug interactions are nowadays an important decision factor in many healthcare interventions. Patients with cardiovascular risk factors such as hyperlipidemia and hypertension are usually prescribed long-term treatments. We need more informed decision tools to direct future clinical research and decision making to avoid HDI occurrences in this group. Methods: A scoping review was conducted using data from online databases such as PUBMED, the National Library of Medicine, and the electronic Medicines Compendium. Included studies consisted of the reported effects on Phase 1/2 and P-glycoprotein of herbal medicines listed in the medicines agencies of Latin America and Europe and drugs used for cardiovascular conditions (statins, diuretics, beta blockers, calcium channel blockers, and ACE inhibitors). The cross tabulation of the results allowed for finding potential HDI. Results and conclusions: as per the preclinical data reviewed here, we encourage more clinical research on whether drugs with apparently very low interaction risk, such as pravastatin, nadolol, and nimodipine/nitrendipine, may help prevent HDI when statins, beta blockers, and calcium channel blockers, respectively, are prescribed for long-term treatments.

## 1. Introduction

Herbal product use is widespread. The World Health Organization estimates that the global yearly market of “traditional” herbal products is around USD 80 billion [1]. Some countries (such as Brazil) and/or entire regions (such as the European Union) have regulated their use as medicines [2]. Herbal medicinal products are medicines with one or more active ingredients that are naturally produced by a plant and should be chemically characterized to satisfy the regulator’s requirements set in either the pharmacopoeia or official monographs [3]. Most active herbal ingredients are usually extracted to facilitate the active compounds’ batch-to-batch consistency and pharmacologically relevant levels. However, sometimes it may consist of raw material with little processing. These products are considered medicinal if approved by medicines agencies and, therefore, can be used by patients to relieve illness [1]. As with any other medicine, some adverse effects may arise from using herbal products in combination with therapeutic drugs for some conditions. As a result, the drug’s or natural product’s pharmacological and/or toxicological effect may shift [4]. Herbal–drug interactions can be differentiated into two classes: pharmacodynamic and pharmacokinetic. Pharmacodynamic interaction results in an increased or decreased pharmacological effect of the drug or the herbal product. Pharmacokinetic interactions occur when a drug or herbal product is able to alter the absorption, distribution, metabolism, and elimination (ADME) of another drug [5].

A common belief among patients and healthcare professionals results in reduced data on the adverse effects and interactions resulting from the common use of herbal products caused by under reporting and the belief that herbal products are harmless. Most herbal medicines are classified as “Over-The-Counter products” to complicate matters further. Thus, the frequency and reasons that people use these herbal products worldwide are still largely unknown. Another confounding factor is that two herbal products with the same herbal active may differ in their strength and bioavailability, resulting in varying therapeutical and toxicological outcomes; thus, regulators cannot easily evaluate and anticipate the scale of potential interactions or adverse effects [4,6].

An example of an herb that can cause a drug interaction is garlic (*Allium sativum*). Garlic is marketed with a variety of claimed health benefits. One of the benefits of garlic consumption is the reduced risk of cancer development, especially in the gastrointestinal system. Garlic has an antithrombotic effect. When taken with warfarin, garlic interacts with warfarin, causing an increased international normalized ratio. This means that the patient’s coagulation may be impaired over a longer period than anticipated, thus increasing their risk of bleeding. Patients with a significant dietary garlic consumption before warfarin therapy must be evaluated [4,6,7].

Healthcare workers are more familiar with drug–drug interactions than herbal ones, which can result in catastrophic outcomes when dealing with a diverse healthcare world. A system that reliably reports problems based on evidence of herbal–drug interactions must be set up to resolve such an issue. That system must also identify the significance of the risk and possibility of interactions. Many herbal product consumers believe they can be used as a replacement for pharmaceutical drug therapy. Given that some herbal products’ data may be lacking, patients relying completely on herbal products may risk an untreated or deteriorating condition. In some cases, patients may experience polypharmacy and consume over-the-counter medication that interacts with their herbal products, leading to detrimental interactions between the substances. Patients may purchase herbal products after obtaining the drug with little or no awareness of the potential of any interaction. Healthcare systems must protect the patients’ vulnerability to such hazards [2,5].

Our research group has a long track record in ascertaining and reviewing the pharmacokinetic effects of herbal medicines [8,9,10,11,12]. Identifying the process and duration of the absorption, distribution, metabolism, and excretion of a herbal drug (xenobiotic) after being administered into the body is key to predicting HDI [13].

The chemical complexity of herbal medicines makes this a challenging task. To start with, the oral absorption of the many active and inactive compounds within may depend on the pH of the environment and the pK_a_ of the herbal ingredient. Transporters such as P-glycoprotein (P-gp) can impede the absorption of xenobiotics by using up the energy and transporting the drug back to the gastrointestinal system [14].

Metabolism follows absorption. It is mainly a biphasic pharmacokinetic stage, producing a compound with different chemical and pharmacological characteristics than its precursor compound. Most metabolism of the orally administered drug occurs in the liver as first-pass metabolism, with the remainder being metabolized by other organs around the body. In Phase 1 of metabolism, the cytochrome P450 isozymes are responsible for changing the compound’s structure by oxidation, reduction, or hydroxylation, mostly by the action of CYP3A4/5 enzymes. Phase 2 uses UDP-glucuronosyltransferase enzymes (UGT) to detoxify the products of Phase 1 via conjugation. Metabolism can alter the structure of a molecule, reducing its function or completely stopping its function. It can also produce a functioning molecule from an inactive compound (such as prodrugs) [15,16,17].

A portion of absorbed compounds that may or may not have undergone structural changes is released from the body [18]. The elimination rate is important as it dictates the time length to reach a therapeutic response, the number of dosing required, and the frequency. Most water-soluble compounds and metabolites are eliminated by the kidneys using glomerular filtration. However, reabsorption prevents lipid-soluble drugs from being eliminated by the kidney. To facilitate such a mechanism, lipid-soluble drugs must be reabsorbed and metabolized into water-soluble compounds. Transporters can excrete acidic and basic compounds in the membranes of the kidneys [15,16].

Despite a shortage of data on the ADME parameters affected by herbal drugs, most national formularies already include mention to some clinically relevant metabolic interactions between herbal supplements and medicines, such as in Appendix 1 of the *British National Formulary* [19]. Most of the available reports deal with their action on cytochromes; only a fraction looks at Phase 2 and important transporters such as P-glycoprotein. Moreover, these data are only discussed in the literature for a few “important” herbal medicines such as St. John’s wort, ginkgo, valerian, etc. We decided to select in this paper several less-known herbal medicines that are in use both in Europe and Latin America. The rationale for the selection (and the ADME parameters of these herbal medicines) is the subject of our previous review [20]. Briefly, it is based on the RENISUS list, which is a positive list of essential herbal medicines approved by the Brazilian Health System for clinical use. The Brazilians include herbal medicines that are also quite popular in the wider Latin American region, thus potentially available to hundreds of millions of persons. As far as we know, this is the first time their potential interaction with common drugs used for cardiovascular indications is attempted.

Our aim is to cross map as many of those parameters for clinically relevant selected herbal medicines (i.e., approved by regulators) and cardiovascular drugs, including statins, diuretics, calcium channel blockers, beta blockers, and ACE inhibitors. The data will be used to cross check and discuss potential herb–drug metabolic interactions between them. Our ultimate objective is to inform further in vivo and/or clinical research.

## 2. Method

The study design is considered a scoping review (data mining) study. Based on publications available from online resources, information is extracted and used to identify the potential for preclinical interactions between natural herbal products and some drugs used for cardiovascular conditions (statins and diuretics). To begin with, some information regarding the pharmacological effect of herbal drugs is extracted from Mazzari et al. [20]. The information mainly concerns the inhibition or induction of natural herbal products on some pharmacokinetic targets in the body. Following that, a list of available drugs that fall into the categories of statins and diuretics is created using the *British National Formulary* [19]. Then, information about the drug ADME is extracted from published online sources using tools such as Google Scholar (scholar.google.com (accessed between November 2020 and January 2021), the National Library of Medicine: National Center for Biotechnology Information (pubmed.nbci.nlm.nih.gov (accessed between November 2020 and January 2021)), the electronic Medicines Compendium (www.medicines.org.uk (accessed between November 2020 and January 2021)), and Google’s search engine (www.google.co.uk (accessed between November 2020 and January 2021)). The searched phrases entered include the drug’s name and are followed by key terms such as: “metabolism”, “metabolite”, “absorption”, “transport”, “pharmacokinetics”, “pharmacology”, “CYP”, “P-gp”, “P-glycoprotein”, “glutathione”, “UGT”, “pharmacokinetic interactions”, “induction”, “inhibition”, “diuretics”, “loop diuretics”, “thiazide diuretics”, “potassium-sparing diuretics”, “osmotic diuretic”, “carbonic anhydrase inhibitor”, “side effects”, “effect”, “toxicity”, “mode of action”, “ADME”, “calcium-channel blockers”, “beta-blockers”, and “ACE inhibitors”. A combination of the drug name followed by the relevant terminology from the terms listed should get the relevant resources. Once the results are discovered, the article covering the topic required is investigated. The cytochrome P450 isoenzymes (or any other enzymes) involved in the drug’s metabolism are noted. A table of active herbal medicinal products of relevance in the Americas and/or Europe (and the enzymes they induce/inhibit) is tabulated against the enzymes involved in the drug’s metabolism.

## 3. Results

Table 1 shows all results obtained covering the following ADME parameters: P-glycoprotein activity for absorption, cytochromes CYP1A2, CYP2C9, CYP2D6, CYP2E1, CYP3A4, CYP3A5, expression, and/or activity for Phase 1 metabolism; UGT activity and glutathione levels for Phase 2 metabolism.

## 4. Discussion

### 4.1. Statins

Statins (a class of lipid-modifying drugs) reduce circulating low-density lipoprotein cholesterol (LDL-C), thus lowering the likelihood of coronary heart disease occurring in patients with elevated LDL-C and patients with normal levels. They work by inhibiting the hydroxymethyl glutaryl-coenzyme A (HMG-CoA) reductase, which promotes mevalonate formation from HMG-CoA. In this way, statins block cholesterol synthesis in the hepatocytes and their intracellular levels. That increases the expression of LDL receptors on the cell membranes of the hepatocytes and results in the absorption of the LDL-C from the general circulation, thus eventually lowering the body’s cholesterol levels as a result [28].

Statins are metabolized by various cytochrome P450 (CYP) isoenzymes. For instance, atorvastatin [28,29,34], fluvastatin [30], and simvastatin [28,29,35] are all metabolized by CYP3A4. Atorvastatin is also metabolized by CYP3A5 [29]. In the case of fluvastatin, other CYP enzymes involved in its metabolism are CYP2C8 [30] and CYP2C9 [28,29,30,31]. Simvastatin is metabolized by CYP2C8 and CYP3A5 [29]. Rosuvastatin is slightly metabolized by CYP2C9 [29,31]. Pravastatin is not metabolized [28,29,40,41].

Many herbal products (Table 1) can induce or inhibit these CYP isoenzymes, which can either increase or decrease the metabolism of most statins, except for pravastatin, which is largely excreted unmetabolized by the kidneys. The concurrent use of herbal medicines inhibiting one or more of the enzymes responsible for the metabolism of the statin will lead to an increased amount of an inactive form of the drug in circulation. The increased bioavailability of the inactive form of the drug may increase the risk of experiencing side effects such as myopathy and rhabdomyolysis whilst not providing a full therapeutic effect [31]. Therefore, statins should not be used with enzyme inhibitors, particularly garlic, turmeric, mint, pomegranate, and clover, that affect most of these isoforms more or less to some extent. As statins seem not to be subjected to significant Phase 2 metabolism [42], then herbal medicines that do not exert an effect on CYPs, such as yarrow, aloe, cashew, carqueja, Brazilian orchid tree/pata de vaca, artichoke, murungu, pepper/rosemary, guaco, guava, and rue, may not interact (at least significantly) with statins.

If the enzyme responsible for the metabolism of the statin is induced, it may result in varying outcomes from one drug to another. For example, in the case of induction in metabolism by CYP 3A4, fluvastatin is converted at a greater amount to its inactive metabolite [28]. This may reduce the therapeutic effect of fluvastatin, making a patient with hypercholesteremia more susceptible to cardiovascular diseases. In the case of atorvastatin and simvastatin, an enzyme inducer causing an increase in their metabolism may result in an increased active metabolite in the general circulation. The lipid profile must be monitored if atorvastatin or simvastatin are prescribed with an inducer of one of the enzymes involved in their metabolism as the enzyme inducers may reduce the bioavailability of atorvastatin and simvastatin.

### 4.2. Diuretics

Diuretics are a class of drugs used to treat hypertension, hypervolemia, heart failure, and electrolyte disorders. The term “diuretics” includes several subgroups that work in different mechanisms but all result in increased urine production and reduced blood volume. There are three main subgroups of diuretics: thiazide and thiazide-like diuretics, loop diuretics, and potassium-sparing diuretics. In some ways, diuretics may replace β-blockers and angiotensin-converting enzyme (ACE) inhibitors in the treatment of hypertension [43].

Thiazide and thiazide-like diuretics affect the distal convoluted tubule. They inhibit the NaCl cotransporters. This reduces the reabsorption of Na^+^ and Cl^−^, which promotes diuresis. They also increase the reabsorption of Ca^2+^ (useful in treating calcium-containing kidney stones) and may cause the excretion of K^+^ [43]. Thiazide and thiazide-like diuretics include bendroflumethiazide, chlortalidone, chlorothiazide, hydrochlorothiazide, hydroflumethiazide, indapamide, metolazone, and xipamide. Bendroflumethiazide is extensively metabolized [44] and a recent report suggests that CYP2C19, CYP2C8, and CYP3A4 are involved in the metabolism of indapamide, but there is limited information regarding its metabolism. On the other hand, chlortalidone, chlorothiazide, hydrochlorothiazide, hydroflumethiazide, and metolazone seem not to be metabolized by Phase 1 enzymes [24]. This would imply that there should not exist any significant interaction between the selected herbal medicines with these drugs at the Phase 1 level. However, indapamide and xipamide are largely excreted as O-glucuronide [38]. Garlic increases the expression of UGTs [20], thus potentially reducing their therapeutic effect, whereas turmeric inhibits the expression of these enzymes in the liver, potentially causing hyponatremia and hypokalemia.

Loop diuretics include bumetanide, furosemide, and torasemide. They inhibit the Na^+^-K^+^-2Cl^−^ cotransporter in the ascending limb of the loop of Henle, therefore the reabsorption of Na^+^, K^+^, and Cl^−^ in the filtrate within the lumen is impeded. This promotes natriuresis and diuresis [43,45]. Bumetanide is metabolized by uridine 5′-diphosphate glucuronosyltransferase [24]. Bumetanide’s metabolites are inactive [46]. Furosemide is mainly excreted unmetabolized via the kidneys [47]. The herbal medicines selected in this study do not metabolically interact with bumetanide and furosemide due to their pharmacokinetics not being impacted by CYP isoenzymes. Torasemide is metabolized by CYP2C8 and CYP2C9 [24] into inactive metabolites [48]. None of the selected herbal medicines were reported to affect CYP2C8, but CYP2C9 activity may be modulated and inhibited by garlic, eucalyptus, devil’s claw, mint, pomegranate, clover, and ginger. Inhibition of CYP2C9 will increase the bioavailability of torasemide, which will increase the risk of adverse effects [49] such as headaches, dizziness, fluid and electrolyte imbalances (such as hypokalemia), and muscle spasms [50]. In the case of garlic, no apparent effect can be derived from the data as they point to also being able to increase the expression levels of the enzyme, which would, in turn, lead to the increased metabolism of torsemide and a reduced therapeutic effect.

The potassium-sparing diuretics are groups of drugs that affect reabsorption in the collecting duct. They can be subcategorized into pteridine analogues (including amiloride and triamterene) and aldosterone receptor antagonists (eplerenone and spironolactone). Pteridine analogues (amiloride and triamterene) have a mechanism of action where they inhibit the reabsorption of Na^+^ via the epithelial Na^+^ channels (ENaC). This, however, is not significant enough for the diuretic effect and some K^+^ is excreted if used in monotherapy. Other drugs can be combined with pteridine analogues to increase the conservation of K^+^ in the body and to prevent hypokalemia. Pteridine analogues can be used with aldosterone receptor antagonists. Aldosterone receptor antagonists function by reducing the activity of the Na^+^/K^+^ pump and the ENaC, which are sensitive to aldosterone, in the basolateral cells. This whole mechanism prevents the reabsorption of Na^+^ and the excretion of K^+^ and H^+^ [43].

Triamterene is metabolized by CYP1A2 and sulfotransferases [24] into its active metabolite [51]. The induction of CYP1A2 (by garlic and/or turmeric) would increase the therapeutic effect and reduce the drug’s half-life and bioavailability, whilst inhibition—for example, by eucalyptus, soya, devil’s claw, mint, bitter melon, gale of the wind, pomegranate, and/or clover—would increase its plasma levels and side effects. Eplerenone is metabolized by CYP3A4 [24,36] and CYP3A5, producing an inactive metabolite [36]. CYP3A4 and CYP3A5 inhibition (pomegranate can simultaneously inhibit both CYPs) would increase eplerenone’s bioavailability, increasing its diuretic effect and risk of adverse effects. The induction of CYP3A4 and CYP3A5 would reduce the therapeutic effect of eplerenone. Contrarily, spironolactone is a prodrug and needs to be metabolized by CYP3A4 [24] to yield active metabolites [36]. Induction of this isoform would result in an increased diuretic effect, whereas the inhibition of spironolactone would cause a decreased diuretic effect and an increased risk of an adverse drug reaction. None of the selected herbal medicines seem to increase CYP3A4 levels, so we should not expect significant interactions in this direction, although other well-known herbs such as St. John’s wort and common valerian strongly induce CYP3A4, whilst Ginkgo biloba increases its activity, as reported elsewhere [25]. Amiloride is not metabolized [24], so the induction or inhibition of CYP isoenzymes do not affect it and would be theoretically exempt from HDIs at this level.

Carbonic anhydrase inhibitors include acetazolamide, brinzolamide, and dorzolamide. They reduce the reabsorption of sodium ions, bicarbonate ions, and water. These ions and water are then transferred to the distal collecting duct to be excreted [43]. Both acetazolamide [52] and dorzolamide [53] are excreted unmetabolized, so enzyme inhibition or induction would not affect bioavailability in terms of metabolism. Brinzolamide is metabolized by CYP3A4 and CYP2C9 [32], which may result in a significant number of interactions with many herbal medicines (see Table 1). The main metabolite of brinzolamide is N-desethyl-brinzolamide, which is an active metabolite [54]. If an enzyme inhibition occurs in CYP3A4 and CYP2C9, an inhibitory effect on the production of N-desethyl-brinzolamide will occur, reducing the effect. However, eliminated N-desethyl-brinzolamide accounts for only 6% of administered brinzolamide; the rest is mainly excreted as unchanged brinzolamide [54]. This means that, although the CYP enzyme inhibition of brinzolamide has some effect on the overall inhibition of carbonic anhydrase, it is not technically severe.

Mannitol is an osmotic diuretic, which means diuresis results from osmosis along the renal tubule. It is excreted via the renal medulla; water is also excreted [43]. Mannitol is not metabolized [55]. As a result, the reviewed herbal products would not exert any metabolic interactions here.

### 4.3. Calcium Channel Blockers

Calcium channel blockers decrease arterial blood pressure by blocking voltage-gated calcium channels to prevent an influx of calcium, relaxing vascular smooth muscle, and lowering systemic vascular resistance. As an entirety, they exhibit extensive first-pass metabolism due to the low and unpredictable oral bioavailability [24]. Calcium-channel blockers can be split into dihydropyridine and non-dihydropyridine [56].

Dihydropyridines work on the smooth muscle, causing vasodilation in the arteries; examples include amlodipine, nifedipine, and felodipine [56]. Amlodipine and nifedipine are extensively metabolized in the liver and act as a substrate for CYP3A4. Felodipine undergoes first-pass metabolism as it is nearly completely absorbed when orally administrated; however, it can also be a substrate for CYP3A4. Herbs that inhibit CYP3A4, such as pomegranate, peppermint, and fennel, will increase these drugs’ levels when taken concurrently. This will raise the risk of adverse effects such as severe abdominal pain, nausea and vomiting, and dysentery. A surge of these drug levels in the body can lead to toxicity, characterized by shortness of breath, sleepiness, and fast heart rate due to low blood pressure [24,25].

Regarding Phase 2, amlodipine and nisoldipine act upon UGT, which can interact with garlic and turmeric. Both garlic and turmeric inhibit UGT, leading to an increased concentration of amlodipine or nisoldipine when co-administered. This can be displayed by signs of adverse effects or, worse, toxicity, hypotension with bradycardia, confusion, PR prolongation, and, ultimately, heart failure [24].

Non-dihydropyridine calcium channel blockers act upon the cardiac tissue, decreasing the heart rate and contractility, ultimately reducing blood pressure. Diltiazem and verapamil are both present in this subclass and both inhibit the action of CYP3A4. These two drugs interact with herbs that act upon CYP3A4 and, as the herbs and these drugs both inhibit this cytochrome, it will increase levels of these drugs in the body. This can lead to toxicity and ultimately it will be extremely difficult for the drug to be excreted. However, diltiazem is an inhibitor and substrate of this enzyme, so it will partly act in a similar way to dihydropyridines; nevertheless, the outcome will be similar to adverse effects and, in the worst case, toxicity. Verapamil also takes action upon many other cytochromes, making this the least desirable calcium channel blocker to use as there is a greater chance for interactions with a larger variety of herbs [24,25].

P-gp inhibitors can increase the bioavailability of xenobiotics. As P-gp is inhibited, more of the drug can penetrate the blood–brain barrier (BBB), leading to CNS toxicity nifedipine and diltiazem are both moderate inhibitors for P-gp, and verapamil has strong inhibitory effects for P-gp [26]. Garlic, yarrow, and turmeric also inhibit P-gp during absorption [57]. Therefore, when co-administered with these calcium channel blockers, there may be adverse effects such as constipation, dizziness, and palpitations, potentially leading to toxicity. In turn, we may experience adverse herb effects such as “turmeric-induced” diarrhea, headache, and skin rashes [58].

Some dihydropyridine calcium channel blockers are known inhibitors of carboxylesterases, including felodipine, nitrendipine, isradipine, and amlodipine, having the highest inhibition potential [59]. Diltiazem and verapamil target carboxylesterases with varying inhibition capabilities. No findings have shown that herbs target carboxylesterases during Phase 1 metabolism, hence it is difficult to ascertain any HDIs.

### 4.4. Beta Blockers

Beta blockers exert their effect in the treatment of hypertension by blocking the effect of adrenaline on the heart to decrease the heart rate and to lower blood pressure [60]. The hepatic metabolism of most beta blockers includes Phase 1, with the addition of Phase 2 glucuronidation reactions to be eliminated by the kidneys. Predominantly, beta blockers act as substrates for CYP2D6, interacting with many herbs that inhibit this isoform and resulting in adverse effects such as nausea, vomiting, and dizziness and, in severe cases, toxic effects such as bradycardia, hypotension, arrhythmias, and seizures [61].

Metabolites of propranolol and betaxolol are produced through reactivity with CYP1A2 and CYP2D6. Betaxolol metabolites catalyzed by these cytochromes have minimal impact on its clinical effect and a very small amount of the drug is excreted in the unchanged form. Certain herbs exhibit an inhibitory effect on these cytochromes, meaning that, when co-administered with these specific beta blockers, adverse effects will likely arise. Garlic and turmeric induce CYP1A2, increasing the rate of metabolism for betaxolol or propranolol when taken concurrently [24]. This can exacerbate hypertension, as the beta blockers are ineffectively treating the condition.

Bisoprolol is metabolized hepatically via oxidation with no conjugation, primarily by CYP3A4, to form inactive metabolites; it acts as a substrate for CYP2D6, which is not clinically significant [25]. Red clover, gale of wind, and grapple plant all inhibit CYP2D6 and CYP3A4; therefore, when taken concurrently, bisoprolol will not be metabolized. Toxicity can arise from increased concentrations of bisoprolol in the body, which can present with bradycardia, fatigue, and hypotension [19].

Four beta blockers have been found to target P-gp: bisoprolol, metoprolol, propranolol, and carvedilol (the most potent inhibitor), whilst atenolol acts as a substrate for P-gp [26]. Garlic, turmeric, and yarrow are the only herbs inhibiting P-gp; hence, when taken, the effects of beta blockers concurrently can enter into the toxic window. Propranolol and carvedilol are highly lipophilic, so they can easily pass the BBB when P-gp is inhibited, triggering seizures with propranolol and carvedilol and instigating a potentially altered mental state [61].

The most favorable beta blockers for treating hypertension in patients utilizing herbal medicines would be nadolol as it is excreted in the unchanged form and, therefore, will not likely interact metabolically with herbal medicines. Carvedilol displays the biggest disadvantage in this patient group as it acts as a substrate for most cytochromes compared with other beta blockers, leading to various herb–drug interactions. This poses a significant risk to patients as there is potential for decreased therapeutic effects, adverse effects, and/or toxicity [24].

Carvedilol is the only beta blocker that targets carboxylesterases with inhibitory effects. All other beta blockers demonstrated no inhibition potential for carboxylesterases [59,62]. The lack of data to display whether herbs target carboxylesterases prevents a determination of whether a herb–drug interaction will occur when administered concomitantly with carvedilol.

### 4.5. ACE Inhibitors

ACE inhibitors are utilized in treating hypertension as they cause vasodilation by preventing the formation of angiotensin II by inhibiting the angiotensin-converting enzyme. Most ACE inhibitors are prodrugs that undergo hydrolysis from an inactive ester to a carboxylic acid that is therapeutically active. The exceptions to this are captopril and lisinopril, which are already active and do not undergo hydrolysis, suggesting ample oral bioavailability [62]. Approximately half of the administered dose of captopril is eliminated in the unchanged form. The remainder undergoes hepatic carboxylation to inactive disulfide metabolites. Lisinopril is the only ACE inhibitor not metabolized and excreted entirely unchanged as it is water soluble [24].

Carboxylesterases (CES) are the main targets for ACE inhibitor metabolism, mostly found in the small intestine and the liver. For substrate drugs, this can significantly lower their bioavailability as they are hydrolyzed by first-pass metabolism [59,62]. Cilazapril, enalapril, moexipril and spirapril are exclusively bioactivated by CES [24]. Trandolapril and ramipril are also prodrug ACE inhibitors that target CES as substrates to form their active metabolites [61]. There are no data on how the selected herbs impact CES activity.

The remaining ACE inhibitors are prodrugs that are bioactivated in Phase 1 as they target CES and then undergo glucuronidation conjugation reactions as they target UGT in Phase 2 [24]. Garlic is an inducer of UGT. When taken concurrently with these ACE inhibitors, the increased efficacy of UGT leads to the faster metabolism of the ACE inhibitors. Ultimately, there would be therapeutic inefficacy of the ACE inhibitor, resulting in the ineffective treatment of hypertension. Garlic has proven to reduce blood pressure in hypertensive patients, working similarly to an ACE inhibitor, blocking angiotensin-II production, therefore promoting vasodilation and thus reducing blood pressure [63]. Hence, despite the interaction between garlic and ACE inhibitors, the patient’s hypertension may not have fatal effects when taken concurrently. Turmeric inhibits UGT, instigating a slower metabolism of ACE inhibitor, resulting in an advanced amount of the drug in the body. This gives the potential for the patient to experience adverse effects such as dry irritant cough, angioedema, and hyperkalemia [64].

Captopril is the only ACE inhibitor that targets P-gp during metabolism with an inhibitory reaction [27]. Garlic, turmeric, and yarrow are also inhibitors of P-gp, so, when taken concurrently with captopril, the rate of metabolism is significantly reduced [20]. Therefore, captopril’s concentration and therapeutic efficacy increase, resulting in adverse effects such as dry persistent cough, skin rash, or dizziness [65].

## 5. Potential Clinical Relevance of the Findings

### 5.1. Statins

With herbal remedies being used very frequently by the general public, many users and prescribers may be unaware of the pharmacokinetics of these herbal products. Some of these preclinical data can be used as guidance for prescribers, pharmacists, and patients who are taking statins for serious cardiovascular conditions. If a patient is given a statin and an herbal product that interacts with the statin they are using, the effects on their health could be detrimental. Enzyme inhibitors can increase the systemic bioavailability of statins and cause the patient to have a higher risk of experiencing serious adverse drug reactions [31]. Depending on the seriousness of that reaction, an increase in hospitalizations or visits to the general practitioner may occur, which can be avoided if both patient and prescriber were provided with further resources to come to a well-informed and mutual decision on a beneficial healthcare plan for a patient with hyperlipidemia.

On the other hand, if the patient were using an enzyme inducer that interacts with a statin, their hyperlipidemia would not resolve due to the increased metabolism of the statin [31]. Therefore, the patient may not report any symptoms of adverse drug reaction for a while, but they may then develop cardiovascular disease due to their untreated or undertreated hyperlipidemia. A good option to avoid any of these problems regarding herbal–drug interactions is to switch the patient to pravastatin. Pravastatin is not metabolized [28,29,40]; this can solve the issue in terms of metabolism as enzyme inducers and inhibitors will not affect the bioavailability of pravastatin, at least in terms of metabolism. Pravastatin is effective in reducing LDL-C [66]. Therefore, switching patients to pravastatin shall not risk compromising the treatment of their hyperlipidemia.

### 5.2. Diuretics

Patients may be using an herbal product simultaneously with diuretics that do not undergo metabolism by CYP isoenzymes, such as amiloride. Although not a pharmacokinetic interaction, it is worth reminding that oral aloe (in particular aloe latex, which also includes anthraquinones) should not be used by patients on diuretics because it can decrease levels of potassium in the body. Similar herb–drug interactions between diuretics and herbal products for other important herbs not reviewed here have been recorded in major publications, such as *Stockley’s Herbal Medicines Interactions* [6].

### 5.3. Antihypertensives

ACE inhibitors have only been found to interact with garlic, turmeric, and yarrow metabolically. Captopril interacted with all three of these herbs; the remaining ACE inhibitors only interacted with garlic and turmeric. More than half of these HDIs could result in toxicity, potentially decreasing their therapeutic efficacy.

Beta blockers can potentially cause many HDIs during metabolism due to corresponding targets with herbal products. Nadolol is excreted in the unchanged form, therefore this would be the most favorable beta blocker for treating hypertension in patients frequently taking herbal products. Among the beta blockers, atenolol had the lowest chance of interacting with herbs, whereas carvedilol interacted with potentially all herbal medicines. Most interactions with this class may lead to toxicity, with a minority capable of increasing metabolic rate causing therapeutic inefficacy.

Calcium channel blockers will probably cause many HDIs owing to the abundance of herbs targeting CYP3A4, which is a key metabolic factor for all calcium channel blockers. Most calcium channel blocker HDIs are theoretically capable of entering the toxic window. A few could lead to an increase in metabolism, putting the patient at risk of unmanaged hypertension. Verapamil, a notable P-glycoprotein inhibitor, is particularly prone to interact with herbs. Nimodipine and nitrendipine had less interaction potential, in any case.

### 5.4. Limitations and Assumptions of This Study

The study’s main limitation was the scarcity of data on metabolic targets for herbs as many interactions may not have been identified. The effect and severity of interactions are unknown when dealing with such preclinical findings, therefore assumptions of adverse effects have to be extrapolated with caution.

A main limitation of the study is that different countries and regions may use different herbal products. Not all the information on the pharmacokinetics of herbal products is discovered regarding bioavailability, distribution, and elimination rates. For herbal products without pharmacokinetic data, patients’ harm may be caused and can go undetected. The frequency of the herbal–drug interactions and whether the interactions are dose-related is largely unknown [67].

To further complicate matters, the metabolic pathway of some drugs, such as bendroflumethiazide [68] is still unknown. In addition, there may be interactions with other enzymes that do not belong to the CYP isoenzymes group. Some drugs are given non-systemically, such as dorzolamide [52], which will need a further investigation about whether or not ingesting a herbal product will affect its localized effect. Human trials on the effect of *Curcuma longa* showed no effect on the interaction on some enzymes that are theoretically thought to be impacted by some herbal products. For example, in human trials, *Curcuma longa* has proven to increase some drugs affected by CYP1A2 but not UGT [69]. The study investigated pharmacokinetic interactions with drugs but did not cover pharmacodynamic interactions. However, the main active principles (curcuminoids) are shown to effectively interact with P-gp and CYPs in the intestine, thus affecting both absorption and metabolism in this organ and significantly influencing plasma levels in experimental animals [38].

After reviewing the pharmacokinetic data given by the articles/journals, a plethora of HDIs were found with antihypertensives. Calcium channel blockers and beta blockers were found to cause the most interactions with herbs compared with ACE inhibitors. Therefore, ACE inhibitors would be the preferred antihypertensive to treat patients currently taking herbal medicines. However, this is theoretical as the prescribed medicine would always prioritize over any herbal medicines taken concurrently.

There is potential for this review to have a global impact as the use of herbal medicines increases worldwide. Therefore, prescribers must be suitably trained to ensure the safe use of herbal and essential medicines. It would be beneficial to continue this research beyond preclinical interpretation to gain insight into the effects of the interactions.

## 6. Conclusions

The main cardiovascular drugs (hypolipidemic and antihypertensive groups) are reviewed here for their preclinical data on ADME targets and cross tabulated alongside a list of “essential” herbal medicines of Brazil and Europe. Typically, healthcare advice on HDI tries to discourage patients from using herbal medicines and certain foods. This may be feasible in short-term therapies but is unrealistic in most long-term or lifetime treatments. Many of the drugs reviewed here for their preclinical evidence to interact with herbs are first-line treatments for chronic cardiovascular conditions such as hyperlipidemia, oedema, and hypertension. It seems to the authors that more emphasis should be put on choosing a drug within each therapeutic class with the minimum interaction with the ADME targets. As per the preclinical data reviewed, we encourage more clinical research on whether drugs such as pravastatin, nadolol, and nimodipine/nitrendipine may help prevent HDI when statins, beta blockers, and calcium channel blockers, respectively, are prescribed for long-term treatments.

## Figures and Tables

**Table 1 plants-12-00623-t001:** Cross map of documented effects of medicinal plant extracts and studied cardiovascular drugs on P-glycoprotein (P-gp), Phase 1 enzymes and Phase 2 targets.

Plant Species (Family) Region/s Common Names (*)	Herbal Extracts [20]	Statins	Diuretics	Calcium Channel Blockers	Beta Blockers	ACE Inhibitors
*Achillea millefollium* (Compositae) Europe/Latin America Yarrow/Milefólio/Milenrama	P-glycoprotein (−)	Atorvastatin [21]	Acetazolamide [22], Spironolactone [23]	Diltiazem, Felodipine, Isradipine, Nifedipine, Verapamil [24,25,26]	[24,27]	Captopril [24,25,27]
*Allium sativum* (Amaryllidaceae) Garlic/Alho/Ajo Europe/Latin America	CYP1A2 (+)		Triamterene [24]	Verapamil [24,25,26]	Carvedilol, Propranolol [20,24,27]	
	CYP2C9 (−,+)	Fluvastatin [28,29,30,31], Rosuvastatin [29,31]	Brinzolamide [32], Torasemide [24], Indapamide [33]	Verapamil [24,25,26]		
	CYP2E1 (−)				Carvedilol [20,24,27]	
	CYP3A4 (NE,−)	Atorvastatin [28,29,34] Fluvastatin [30] Simvastatin [28,29,35]	Brinzolamide [32] Eplerenone [24,36] Spironolactone [24] Indapamide [24]	Amlodipine, Diltiazem, Felodipine, Isradipine, Nifedipine, Nimodipine, Nitrendipine, Nisoldipine, Verapamil [24,25,26]	Betaxolol, Bisoprolol, Carvedilol, [20,24,27]	
	CYP3A5 (NE,−)	Atorvastatin [29] Simvastatin [29]	Eplerenone [36]			
	Glutathione (+)		Indapamide [24]			
	UGT (+)		Bumetanide [24], Furosemide [37], Xipamide [38]	Amlodipine [24,25,26]		Benazepril, Fosinopril, Perindopril, Quinapril, Ramipril, Trandolapril [24,25,27]
	P-glycoprotein (−)	Atorvastatin [21]	Acetazolamide [22] Spironolactone [23]	Diltiazem Felodipine, Isradipine, Nifedipine, Verapamil [24,25,26]		Captopril [24,25,27]
*Aloe vera/Aloe* barbadensis (Xanthorrhoeaceae) Europe/Latin America Aloe/Aloe/Áloe	Glutathione (−,+)		Indapamide [24]			
*Anacardium occidentale* (Anacardiaceae) Cashew/Caju/Anacardo Latin America	Glutathione (+)		Indapamide [24]			
*Baccharis trimera* (Compositae) Latin America -/Carqueja/-	Glutathione (−)		Indapamide [24]			
*Bauhinia forficata* (Leguminosae) Latin America Brazilian orchid tree/pata de vaca/pezuña de vaca	Glutathione (−)		Indapamide [24]			
*Chamomilla recutita* (syn. *Matricaria chamomilla*) (Compositae) Europe/Latin America German chamomile/Camomila/Camomila	Glutathione (+)		Indapamide [24]			
	CYP3A4 (−)	Atorvastatin [28,29,34], Fluvastatin [30], Simvastatin [28,29,35]	Brinzolamide [32], Eplerenone [24,36], Spironolactone [24], Indapamide [24]	Amlodipine, Diltiazem, Felodipine, Isradipine, Nifedipine, Nimodipine, Nitrendipine, Nisoldipine, Verapamil [24,25,26]	Betaxolol, Bisoprolol, Carvedilol [20,24,27]	
	Glutathione (−)		Indapamide [24]			
*Croton cajucara* (Euphorbiaceae) Latin America -/Sacaca/-	Glutathione (+)		Indapamide [24]			
*Curcuma longa* (Zingiberaceae) Europe/Latin America Turmeric/Cúrcuma/Cúrcuma	CYP1A2 (+)		Triamterene [24]	Verapamil [24,25,26]	Carvedilol, Propranolol [20,24,27]	
	CYP3A4 (- first pass) [39]	Atorvastatin [28,29,34], Fluvastatin [30], Simvastatin [28,29,35]	Brinzolamide [32], Eplerenone [24,36], Spironolactone [24], Indapamide [24]	Amlodipine, Diltiazem, Felodipine, Isradipine, Nifedipine, Nimodipine, Nitrendipine, Nisoldipine, Verapamil [24,25,26]	Betaxolol, Bisoprolol, Carvedilol [20,24,27]	
	Glutathione (+)		Indapamide [24]			
	UGT (−)		Bumetanide [24], Furosemide [37], Xipamide [38]	Amlodipine [24,25,26]		Benazepril, Fosinopril, Perindopril, Quinapril, Ramipril, Trandolapril [24,25,27]
	P-glycoprotein [39]	Atorvastatin [21]	Acetazolamide [22], Spironolactone [23]	Diltiazem, Felodipine, Isradipine, Nifedipine, Verapamil [24,25,26]		Captopril [24,25,27]
*Cynara scolymus* (Compositae) Europe/Latin America Artichoke/Alcachofra/Alcachofa	Glutathione (+,NE)		Indapamide [24]			
*Erythrina mulungu* (Leguminosae) Latin America Mulungu/Murungu/Mulungu	Glutathione (+)		Indapamide [24]			
*Eucalyptus globulus* (Myrtaceae) Europe/Latin America Eucalyptus/Eucalipto/Eucalipto	CYP1A2 (−)		Triamterene [24]	Verapamil [24,25,26]	Carvedilol, Propranolol [20,24,27]	
	CYP2C9 (−)	Fluvastatin [28,29,30,31], Rosuvastatin [29,31]	Brinzolamide [32], Torasemide [24], Indapamide [33]	Verapamil [24,25,26]	Carvedilol [20,24,27]	
	CYP2D6 (−)				Acebutolol, Alprenolol, Betaxolol, Bisoprolol, Carvedilol, Metoprolol, Nebivolol, Oxprenolol, Propranolol [20,24,27]	
*Foeniculum vulgare* (Apiaceae) Europe/Latin America Fennel/Funcho/Hinojo	CYP3A4 (−)	Atorvastatin [28,29,34], Fluvastatin [30], Simvastatin [28,29,35]	Brinzolamide [32], Eplerenone [24,36], Spironolactone [24], Indapamide [24]	Amlodipine, Diltiazem, Felodipine, Isradipine, Nifedipine, Nimodipine, Nitrendipine, Nisoldipine, Verapamil [24,25,26]	Betaxolol, Bisoprolol, Carvedilol [20,24,27]	
	Glutathione (+)		Indapamide [24]			
*Glycine max* (Leguminosae) Europe/Latin America Soy/Soja/Soja	CYP1A2 (−)		Triamterene [24]	Verapamil [24,25,26]	Carvedilol, Propranolol [20,24,27]	
	Glutathione (+)		Indapamide [24]			
*Harpagophytum procumbens* (Pedaliaceae) Europe/Latin America Devil’s claw/Unha do diabo/Harpagófito	CYP1A2 (NE,−)		Triamterene [24]	Verapamil [24,25,26]	Carvedilol, Propranolol [20,24,27]	
	CYP2C9 (NE,−)	Fluvastatin [28,29,30,31], Rosuvastatin [29,31]	Brinzolamide [32], Torasemide [24], Indapamide [33]	Verapamil [24,25,26]	Carvedilol [20,24,27]	
	CYP2D6 (NE,−)				Acebutolol, Alprenolol, Betaxolol, Bisoprolol, Carvedilol, Metoprolol, Nebivolol, Oxprenolol, Propranolol [20,24,27]	
*Lippia Sidoides* (Verbenaceae) Latin America Pepper-rosmarin/Alecrim-pimenta/Verbena	CYP3A4 (−)	Atorvastatin [28,29,34], Fluvastatin [30], Simvastatin [28,29,35]	Brinzolamide [32], Eplerenone [24,36], Spironolactone [24], Indapamide [24]	Amlodipine, Diltiazem, Felodipine, Isradipine, Nifedipine, Nimodipine, Nitrendipine, Nisoldipine, Verapamil [24,25,26]	Betaxolol, Bisoprolol, Carvedilol [20,24,27]	
	Glutathione (−)		Indapamide [24]			
*Mentha x piperita* (Lamiaceae) Europe/Latin America Mint/Menta/Menta	CYP1A2 (−)		Triamterene [24]	Verapamil [24,25,26]	Carvedilol, Propranolol [20,24,27]	
	CYP2C9 (−)	Fluvastatin [28,29,30,31], Rosuvastatin [29,31]	Brinzolamide [32], Torasemide [24], Indapamide [33]	Verapamil [24,25,26]	Carvedilol, [20,24,27]	
	CYP2D6 (−)				Acebutolol, Alprenolol, Betaxolol, Bisoprolol, Carvedilol, Metoprolol, Nebivolol, Oxprenolol, Propranolol [20,24,27]	
	CYP3A4 (−)	Atorvastatin [28,29,34], Fluvastatin [30], Simvastatin [28,29,35]	Brinzolamide [32], Eplerenone [24,36], Spironolactone [24], Indapamide [24]	Amlodipine, Diltiazem, Felodipine, Isradipine, Nifedipine, Nimodipine, Nitrendipine, Nisoldipine, Verapamil [24,25,26]	Betaxolol, Bisoprolol, Carvedilol [20,24,27]	
	Glutathione (+)		Indapamide [24]			
*Mentha pulegium* (Lamiaceae) Europe/Latin America Pennyroyal/hortelã-dos-Açores/Poleo	Glutathione (+)		Indapamide [24]			
*Mikania glomerata* (Asteraceae) Latin America Guaco/Guaco/Huaco	Glutathione (+)		Indapamide [24]			
*Momordica charantia* (Cucurbitaceae) Latin America Bitter melon/Melão-de-são-caetano/melón amargo	CYP2E1 (−)				Carvedilol [20,24,27]	
	CYP3A4 (−)	Atorvastatin [28,29,34], Fluvastatin [30], Simvastatin [28,29,35]	Brinzolamide [32], Eplerenone [24,36], Spironolactone [24], Indapamide [24]	Amlodipine, Diltiazem, Felodipine, Isradipine, Nifedipine, Nimodipine, Nitrendipine, Nisoldipine, Verapamil [24,25,26]	Betaxolol, Bisoprolol, Carvedilol [20,24,27]	
	Glutathione (+)					
*Phyllanthus amarus* (Phyllanthaceae) Latin America Stonebreaker/Quebra-pedra/Chancapiedra	CYP1A2 (−)		Triamterene [24]	Verapamil [24,25,26]	Carvedilol, Propranolol [20,24,27]	
	CYP2D6 (−)				Acebutolol, Alprenolol, Betaxolol, Bisoprolol, Carvedilol, Metoprolol, Nebivolol, Oxprenolol, Propranolol [20,24,27]	
	CYP2E1 (−)				Carvedilol [20,24,27]	
	CYP3A4 (−)	Atorvastatin [28,29,34], Fluvastatin [30], Simvastatin [28,29,35]	Brinzolamide [32], Eplerenone [24,36], Spironolactone [24], Indapamide [24]	Amlodipine, Diltiazem, Felodipine, Isradipine, Nifedipine, Nimodipine, Nitrendipine, Nisoldipine, Verapamil [24,25,26]	Betaxolol, Bisoprolol, Carvedilol [20,24,27]	
	CYP3A5 (−)	Atorvastatin [29], Simvastatin [29]	Eplerenone [36]			
	Glutathione (+)		Indapamide [24]			
*Psidium guajava* (Myrtaceae) Latin America Common guava/Guava/Guayaba	Glutathione (+)		Indapamide [24]			
*Punica granatum* (Lythraceae) Europe/Latin America Pomegranate/Romã/Granada	CYP1A2 (−)		Triamterene [24]	Verapamil [24,25,26]	Carvedilol, Propranolol [20,24,27]	
	CYP2C9 (−)	Fluvastatin [28,29,30,31], Rosuvastatin [29,31]	Brinzolamide [32], Torasemide [24], Indapamide [33]	Verapamil [24,25,26]	Carvedilol [20,24,27]	
	CYP2D6 (−)				Acebutolol, Alprenolol, Betaxolol, Bisoprolol, Carvedilol, Metoprolol, Nebivolol, Oxprenolol, Propranolol [20,24,27]	
	CYP2E1 (−)				Carvedilol [20,24,27]	
	CYP3A4 (−)	Atorvastatin [28,29,34], Fluvastatin [30], Simvastatin [28,29,35]	Brinzolamide [32], Eplerenone [24,36], Spironolactone [24], Indapamide [24]	Amlodipine, Diltiazem, Felodipine, Isradipine, Nifedipine, Nimodipine, Nitrendipine, Nisoldipine, Verapamil [24,25,26]	Betaxolol, Bisoprolol, Carvedilol [20,24,27]	
	CYP3A5 (−)	Atorvastatin [29], Simvastatin [29]	Eplerenone [36]			
	Glutathione (+,−)		Indapamide [24]			
*Ruta graveolens* (Rutaceae) Europe/Latin America Rue/Arruda/Ruda	Glutathione (+)		Indapamide [24]			
*Solanum paniculatum* (Solanaceae) Latin America Jurubeba/Jurubeba/Hierba mora	CYP3A4 (−)	Atorvastatin [28,29,34], Fluvastatin [30], Simvastatin [28,29,35]	Brinzolamide [32], Eplerenone [24,36], Spironolactone [24], Indapamide [24]	Amlodipine, Diltiazem, Felodipine, Isradipine, Nifedipine, Nimodipine, Nitrendipine, Nisoldipine, Verapamil [24,25,26]	Betaxolol, Bisoprolol, Carvedilol [20,24,27]	
	Glutathione (−)		Indapamide [24]			
*Trifolium pratense* (Leguminosae) Europe/Latin America Clover/Trevo/Trebol	CYP1A2 (−)		Triamterene [24]	Verapamil [24,25,26]	Carvedilol, Propranolol [20,24,27]	
	CYP2C9 (−)	Fluvastatin [28,29,30,31], Rosuvastatin [29,31]	Brinzolamide [32], Torasemide [24], Indapamide [33]	Verapamil [24,25,26]	Carvedilol [20,24,27]	
	CYP2D6 (−)				Acebutolol, Alprenolol, Betaxolol, Bisoprolol, Carvedilol, Metoprolol, Nebivolol, Oxprenolol, Propranolol [20,24,27]	
	CYP3A4 (−)	Atorvastatin [28,29,34], Fluvastatin [30], Simvastatin [28,29,35]	Brinzolamide [32], Eplerenone [24,36], Spironolactone [24], Indapamide [24]	Amlodipine, Diltiazem, Felodipine, Isradipine, Nifedipine, Nimodipine, Nitrendipine, Nisoldipine, Verapamil [24,25,26]	Betaxolol, Bisoprolol, Carvedilol [20,24,27]	
*Uncaria tomentosa* (Rubiaceae) Europe/Latin America Cat’s claw/Unha-de-gato/Uña de gato	CYP3A4 (−)	Atorvastatin [28,29,34], Fluvastatin [30], Simvastatin [28,29,35]	Brinzolamide [32], Eplerenone [24,36], Spironolactone [24], Indapamide [24]	Amlodipine, Diltiazem, Felodipine, Isradipine, Nifedipine, Nimodipine, Nitrendipine, Nisoldipine, Verapamil [24,25,26]	Betaxolol, Bisoprolol, Carvedilol [20,24,27]	
*Zingiber officinale* (Zingiberaceae) Europe/Latin America Ginger/Gengibre/Gengibre	CYP2C9 (−)	Fluvastatin [28,29,30,31], Rosuvastatin [29,31]	Brinzolamide [32], Torasemide [24], Indapamide [33]	Verapamil [24,25,26]	Carvedilol [20,24,27]	
	CYP3A4 (- first pass) [39]	Atorvastatin [28,29,34], Fluvastatin [30], Simvastatin [28,29,35]	Brinzolamide [32], Eplerenone [24,36], Spironolactone [24], Indapamide [24]	Amlodipine, Diltiazem, Felodipine, Isradipine, Nifedipine, Nimodipine, Nitrendipine, Nisoldipine, Verapamil [24,25,26]	Betaxolol, Bisoprolol, Carvedilol [20,24,27]	
	Glutathione (+,NE)		Indapamide [24]			

(*) English/Portuguese/Spanish; +—enzyme induction; −—enzyme inhibition; NE—no effect.

## Data Availability

No new data were created.

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
