# Peer review of "Potential Pharmacokinetic Interactions of Common Cardiovascular Drugs and Selected European and Latin American Herbal Medicines: A Scoping Review"

_plants, 2023, doi:10.3390/plants12030623_

Round 1
Reviewer 1 Report
The manuscript titled A systematic review of Phase I and II preclinical data of clinically relevant European and Latin American herbal medicines to predict interactions with Cardio-vascular Drugs presents a well-documented pharmacokinetic profile of most widely used cardiovascular drug classes, as well as medicinal plants that interfere with ADME parameters. This paper undoubtedly has its merits, but the authors should make some major changes for the sake of clarity and thus better fitting this Journal’s aims and scope.
My major concerns are the following:
- The authors should clearly state the objectives and the novelty of this manuscript.
- Rewrite the Methods phrase in the Abstract for clarity.
- The definition of pharmacokinetic interactions in the Introduction should be more specific: the authors should mention that the activity of the enzymes and transporters involved in the drug/herb metabolism is altered.
- In the Introduction, the authors state that “Most metabolism of the orally administered drug occurs in the lung,” which is woefully untrue; please correct this phrase (carefully read the cited paper!)
- The authors should clarify the term preclinical; they mention the preclinical effects of drugs and herbs on some enzymes or targets, but most of the drug effects shown in Table 1 are clinical.
- The Discussion section comprehensively presents the pharmacokinetic profile of the cardiovascular drugs of interest within this manuscript. This section is very detailed for drug classes of interest and contains no data about the herbs. Therefore, I recommend the authors discuss some examples of plants or plant extracts known to interfere with enzymes and transporters involved in phase I and phase II cardiovascular drug metabolism.
- Section 5 of this review manuscript titled “… to predict interactions with cardiovascular drugs” must include discussions of herb-drug interactions for each drug class. I recommend the authors present some relevant herb-cardiovascular drug interactions; the authors could also point out the differences in ADR intensity in the combination herb/specific cardiovascular drug. The authors can hypothesize based on Table 1 and also provide some relevant known examples (reported by the literature or online drug-food/herb interaction alert tools, such as DrugBank online interaction checker).
- As the authors introduce in their manuscript title two distinct topological regions with different flora, they should mention the natural habitat of each plant (an extra row in Table 1, the first column)
- This paper deals with drug-herbs interactions. Therefore, I suggest the authors indicate the forms in which plants are processed (from dried plant parts for infusion to pharmaceutical formulations such as capsules or tinctures for oral administration or ointments or gels for topical administration—if this is the case, the absorption rate should be discussed).
Some minor observations:
· please revise some grammar issues (for example, in Introduction: “Most active herbal ingredient are usually extracted”)
· Keep consistency when writing drug names (with or without capital letters along the manuscript)
Author Response
We thank you so much for the helpful comments. It took a few weeks to satisfy them as we were going through the festive season followed by exams season. We appreciate your patience in getting back to you and for the generous extension of the editors in the correction period. The manuscript is vastly improved now and we hope you will be pleased with the outcome. Now in detail, we present below our answers to each of your specific comments.
- The authors should clearly state the objectives and the novelty of this manuscript.
We elaborated on our objectives and novelty in the last paragraph of the introduction (p3, in red)
- Rewrite the Methods phrase in the Abstract for clarity.
Done (in red, p1)
- The definition of pharmacokinetic interactions in the Introduction should be more specific: the authors should mention that the activity of the enzymes and transporters involved in the drug/herb metabolism is altered.
Done (in red, p2)
- In the Introduction, the authors state that “Most metabolism of the orally administered drug occurs in the lung,” which is woefully untrue; please correct this phrase (carefully read the cited paper!)
Thanks for spotting this horrible typo/mistake! It is now corrected with our apologies.
- The authors should clarify the term preclinical; they mention the preclinical effects of drugs and herbs on some enzymes or targets, but most of the drug effects shown in Table 1 are clinical.
Your comment is very interesting. It was extremely difficult to find out what is the exact proportion of pre- and clinical data managed here. Most of our references are either medical databases, books or reviews which are in turn based on a myriad of other references of mixed experimental design. Our feeling is that we surprisingly move in a field managing a majority of pre-clinical (experimental) data. Therefore we decided to change the title and in general the text to a more neutral form to avoid confusion as to what to expect in terms of clinical impact.
- The Discussion section comprehensively presents the pharmacokinetic profile of the cardiovascular drugs of interest within this manuscript. This section is very detailed for drug classes of interest and contains no data about the herbs. Therefore, I recommend the authors discuss some examples of plants or plant extracts known to interfere with enzymes and transporters involved in phase I and phase II cardiovascular drug metabolism.
We have ensured that every section contains such relevant examples. They are in red throughout the discussion section.
- Section 5 of this review manuscript titled “… to predict interactions with cardiovascular drugs” must include discussions of herb-drug interactions for each drug class. I recommend the authors present some relevant herb-cardiovascular drug interactions; the authors could also point out the differences in ADR intensity in the combination herb/specific cardiovascular drug. The authors can hypothesize based on Table 1 and also provide some relevant known examples (reported by the literature or online drug-food/herb interaction alert tools, such as DrugBank online interaction checker).
We changed the title to take out this noun which may be too misleading. Now this section (Potential clinical relevance of the findings) can focus on trying to provide clinical advice in terms of which CDV drugs may be less prone to HDI. The examples of interactions are restricted to the discussion as they are very theoretical. We could not find locate case-reports in the literature and in Stockley’s HDI there is only a vague mention of plants that contain flavonoids to be more prone to interactions. This could not be translated into specific comments as they are very ubiquitous metabolites which quantity vastly differs seasonally and geographically.
- As the authors introduce in their manuscript title two distinct topological regions with different flora, they should mention the natural habitat of each plant (an extra row in Table 1, the first column)
Done (in red) plus we tried to provide all common names in three languages (English, Brazilian Portuguese and Spanish).
- This paper deals with drug-herbs interactions. Therefore, I suggest the authors indicate the forms in which plants are processed (from dried plant parts for infusion to pharmaceutical formulations such as capsules or tinctures for oral administration or ointments or gels for topical administration—if this is the case, the absorption rate should be discussed).
Thanks for rising this comment with which we totally agree but did not see in the beginning. We took out herbs that are indicated for topical use only (such as Calendula officinalis and Cordia verbenacea). In Brazil Aloe and Lippia Sidoides are allowed externally only, but there are many regions where they are consumed orally (off-label or traditional use) and therefore we decided to leave them together with all the rest that are taken orally only.
Some minor observations:
- Please revise some grammar issues (for example, in Introduction: “Most active herbal ingredient are usually extracted”)
- Keep consistency when writing drug names (with or without capital letters along the manuscript)
We have taken care to polish up all the text according to your advice.

Reviewer 2 Report
The issue of drug-herb interaction poses an important safety issue. This review is helpful for practioner to spot the interaction between herbs and cardiovascular drugs. However, I don't think it fits with the criteria of systematic review of PRISMA. It will be more appropriate to call a Scoping Review.
The authors should indicate whether the drug-herb interaction data are in vitro, animal models or human studies. Sometimes the drug interaction are quite strong in in vitro models but sometimes the effects appear only the in vitro model but are unremarkable in human.
Author Response
Dear Reviewer 2
We thank you so much for the helpful comments. It took a few weeks to satisfy them as we were going through the festive season followed by exams season. We appreciate your patience in getting back to you and for the generous extension of the editors in the correction period. The manuscript is vastly improved now, and we hope you will be pleased with the outcome. Now in detail, we present below our answers to each of your specific comments.
- The issue of drug-herb interaction poses an important safety issue. This review is helpful for practioner to spot the interaction between herbs and cardiovascular drugs. However, I don't think it fits with the criteria of systematic review of PRISMA. It will be more appropriate to call a Scoping Review.
Thanks for this helpful comment. We totally see it and agree. We changed accordingly the manuscript.
- The authors should indicate whether the drug-herb interaction data are in vitro, animal models or human studies. Sometimes the drug interaction are quite strong in in vitro models but sometimes the effects appear only the in vitro model but are unremarkable in human.
We also agree in that the experimental model/level of evidence is an important detail. However we soon found that most of our references -that are either medical databases, books or reviews- are in turn based on a myriad of other references of mixed experimental design. We therefore regret not being able to satisfy this request.
Round 2
Reviewer 1 Report
The authors responded properly to all my comments.
I believe the paper is suitable for publication.